# HIV testing and its associated factors among female sex workers in Iran in 2020: Finding from a respondent-driven sampling survey

Fatemeh Tavakoli[1], Ghobad Moradi[2]*, Ali Mirzazadeh[1,3], Bushra Zarei[2], Hamid Sharifi[1]*

1 HIV/STI Surveillance Research Center and WHO Collaborating Center for HIV Surveillance, Institute for Futures Studies in Health, Kerman University of Medical Sciences, Kerman, Iran, 2 Social Determinants of Health Research Center, Research Institute for Health Development, Kurdistan University of Medical Sciences, Sanandaj, Iran, 3 Department of Epidemiology and Biostatistics, Institute for Global Health Sciences, University of California San Francisco, San Francisco, CA, United States of America

* hsharifi@kmu.ac.ir (HS); moradi_gh@yahoo.com (GM)

## Abstract

HIV testing uptake was around 70% among female sex workers (FSWs) in Iran in 2015. Due to the recognized importance of HIV testing in prevention, care, and treatment among FSWs, this study aimed to provide an update and also an improvement as it uses respondent-driven sampling (RDS) for the frequency and the correlates of HIV testing among FSWs in Iran. A total of 1,515 FSWs were recruited from eight cities in Iran between 2019–2020 using RDS. Data were collected using face-to-face interviews. Also, rapid diagnostic tests (RDT) were used to determine HIV and syphilis status as per national guidelines. We used multivariable logistic regression to explore correlates of HIV testing in the last 12 months. Of 1,399 FSWs who had data for HIV testing, 44.7% (95% Confidence Intervals (CI): 40.7; 48.7) reported HIV testing in the last 12 months. The odds of HIV testing in the last 12 months was higher among FSWs who used a condom at last sex with a non-paying partner (Adjusted OR = 1.78; 95% CI: 1.39; 2.27), never used alcohol (Adjusted OR = 1.72; 95% CI: 1.33; 2.23), ever used drug (Adjusted OR = 1.60; 95% CI: 1.22; 2.08). HIV testing among FSWs in Iran is considerably low. It is necessary to know the barriers to HIV testing to improve the cascade of HIV treatment in the country.

## Introduction

The diagnosis UNAIDS 95-95-95 targets calls for actions to ensure 95% of all people living with HIV know their HIV status by 2030 [1]. It looks like it is an ambitious target, as out of 33 million people living with HIV (PLHIV), 19 million (around 57%) do not know their status. This is mainly due to low HIV testing, especially among key populations [2]. In the middle east and North Africa (MENA), only 37% of people living with HIV were diagnosed in 2016 [3].

Female sex workers (FSWs) are one of the key underserved populations at high risk for HIV globally [4]. In Iran, social and cultural sensitivities around FSWs are mainly rooted in the sociocultural, religious, and political context of the country as a Muslim majority setting

Data Availability Statement: Data are owned by the Ministry of Health of Iran and are available from the HIV/STI office located in the Ministry of Health of Iran (E-mail: aids@behdasht.gov.ir; Tel: +98(0)

21-81455055) for researchers who meet the criteria for access to confidential data. Female Sex workers are a highly stigmatized population and sex work is illegal in Iran. To protect the study population, all individual-level data are considered sensitive data. It requires that all researchers who want to work on this data submit their request to the Ministry of Health.

**Funding:** One of our co-authors (Ali Mirzazadeh) has received funding from NIMH (R25MH123256). The funders had no role in study design, data collection and analysis, decision to publish, or preparation of the manuscript.

**Competing interests:** The authors have declared that no competing interests exist.

[5]. Population size estimations suggested that a considerable number of FSWs live and work in Iran [6]. The result of a recent systematic review and meta-analysis in Iran indicated that the pooled prevalence of HIV was 2.2% among FSWs [7]. Also, the findings of two nationwide integrated bio-behavioral surveillance surveys (IBBSS) in 2010 and 2015 reported an overall HIV prevalence of 4.5% and 2.1% among Iranian FSWs, respectively [8].

There is a big gap between the current HIV diagnosis rates in Iran [4, 9, 10] and the UNAIDS 95-95-95 targets for HIV diagnosis [9]. HIV testing can be associated with HIV risk perception, awareness of HIV test, age at sex work initiation, receiving free-of-charge harm reduction services, unstable housing, drug use, the level of income, incarceration history, having paying and non-paying partners, and knowing an HIV testing site [4, 9–11]. The HIV testing uptake was 70.1% among (FSWs) in Iran in 2015. In the current study, we aimed to provide an update on the frequency and the correlates of HIV testing among FSWs in Iran using a nationwide survey conducted in 2020. Also, this study could provide a better picture of HIV testing among FSWs in Iran as this is the first national study that used respondent-driven sampling (RDS) to recruit the participants.

## Methods

### Study design and setting

**Study sampling.** We recruited 1,515 FSWs from geographically diverse eight cities in Iran, namely Bandar-Abbas, Kermanshah, Khorramabad, Mashhad, Sari, Shiraz, Tehran, and Tabriz between December 2019 and August 2020 by using RDS. The cities were selected according to the maximum cultural and geographical variation by the study investigators following consultations with the Ministry of Health and to represent different regions across the country.

**Seed selection.** Sampling started with a non-random selection of seeds. Seeds were selected from different networks based on several criteria: age, geographic region, and risk characteristics related to the subgroup. Finally, 45 seeds were selected, with a minimum of four and a maximum of nine seeds for each city. One of the seeds was non-generative and did not recruit anyone to the study, so she was excluded from the study (44 seeds). Also, a formative assessment was conducted based on in-depth interviews and focus group discussions with key target population members at the beginning of the study.

**Recruitment and eligibility criteria.** The eligibility criteria included: female sex, self-reported sexual intercourse (vaginal or anal) in exchange for money, or any other services with more than one male client in the last 12 months, ≥16 years of age, residing or working in the city of the study at least 12 months before the interview, having a valid RDS coupon (excluding seeds) and consent to participate in this study. For the current analysis, participants were excluded if they were HIV-positive, had unknown HIV status, or did not report their HIV testing history or the time of HIV testing. The details of design, sampling, and data collection approaches have been described elsewhere [12].

### Data collection

Data were collected using face-to-face interviews at RDS centers. After completion of the interview on the behavioral questions, rapid diagnostic tests (RDT) were collected to determine HIV and syphilis status as per national guidelines. HIV and syphilis testing consisted of two steps: a rapid test (SD BIOLINE HIV/Syphilis Duo Rapid Test, Standard Diagnostics, Gyeonggi-do, South Korea) and, if the first rapid test was reactive, confirmation by a second test, i.e., enzyme-linked immunosorbent assay (ELISA).

**Outcome variable: HIV testing in the last 12 months.** FSWs with a lifetime history of HIV testing were asked about the date of their last HIV test followed by a question on whether they knew their test results. The primary outcome, HIV testing in the last 12 months, was treated as a binary variable: those who reported HIV testing in the last 12 months and knew their test result (i.e., had an HIV test result in the last 12 months; code = 1) versus those who had never tested for HIV or had tested for HIV but not in the last 12 months, or had tested for HIV in the last 12 months but did not know the test results (i.e., did not have an HIV test result in the last 12 months; code = 0). Also, we reported common reasons for not taking HIV testing.

**Independent variables.** Data were collected on demographic and baseline characteristics including age (continuous), current marital status (single vs. married vs. divorced, widowed, or sigheh (temporary marriage; which is a type of marriage with a flexible but predetermined duration), educational level (illiterate or primary school vs. secondary or high school vs. college education), monthly income (continuous), age at first sex (<18 vs. ≥18), age at initiation of sex work (<18 vs. ≥18), age at initiation of drug use (<18 vs. ≥18), history of injection drug use (ever) (yes vs. no), history of group sex in the last 12 months (yes vs. no), lifetime abortion (yes vs. no), condom use at last sex with a paying partner (yes vs. no), condom use at last sex with a non-paying partner (yes vs. no), history of the previous incarceration in the last 12 months (yes vs. no), lifetime history of alcohol consumption (yes vs. no), lifetime history of drug use (yes vs. no), lifetime receiving methadone maintenance therapy (MMT) (yes vs. no), HIV knowledge (sufficient vs. insufficient) [Ten questions were asked about HIV transmission; correct responses to all questions were coded as sufficient and otherwise as insufficient], and self-perceived risk of HIV (yes vs. no).

## Statistical analysis

Descriptive statistics, including frequencies, percentages, and 95% confidence intervals (95% CI), were reported for HIV testing in the last 12 months. We reported both RDS-weighted and unweighted prevalence. The prevalence of HIV testing in the last 12 months also was reported for each city. Bivariable and multivariable logistic regression models were fitted to compare the probability of having an HIV test in the last 12 months among different subgroups of FSWs. Variables with a p-value less than <0.2 in the bivariable logistic regression model were entered into the multivariable logistic regression model. The final model was chosen through the backward elimination method using Wald statistics. Data were analyzed using Stata v.14 [13].

Given the lack of consensus on the validity of weighted regression models, unweighted regression models were performed to avoid error rate, have better coverage, increase accuracy, and avoid biased results arising from the RDS weighted analyses [14]. The RDS unweighted regression has been supported by the growing body of literature [15, 16]. Despite this, we also reported RDS-adjusted estimates for HIV testing in the last 12 months by different subgroups of demographics and risk behaviors, which considered network size and homophily within networks. Weighting was done according to Giles'SS estimator and the initial population estimate was considered to be 90,000. RDS-adjusted estimates of HIV prevalence were calculated in RDS-Analyst [17].

## Ethical considerations

All procedures performed in the study conformed to the ethical standards of the Kurdistan University of Medical Sciences Committee (approval ID = IR.MUK.REC.1398.132). The study was anonymous; all women were informed about the study and asked for written informed consent to participate.

### Informed consent

In this study, the informed consent form provided information about the survey to participants and invites them to give informed consent to the survey interview, testing for HIV, and other biomarker tests. The informed consent process started with a short explanation of the survey's purpose, procedures, and potential risks and benefits of participation. The interviewer presented a paper copy of the informed consent form for participants. If the participant could not read or need help understanding the form, the interviewer read the informed consent form to the participant. The interviewer also explained during this session, the anonymous nature of the data collection and the use of consent to ensure anonymity. Once assured that a participant has understood, the interviewer asked the participant to complete the informed consent form and sign it.

### Inclusivity in global research

Additional information regarding the ethical, cultural, and scientific considerations specific to inclusivity in global research is included in the S1 File.

## Results

Out of 1515 participants, 1399 FSWs were included in this analysis. Among excluded participants, 22 FSWs had HIV-positive test results before this study, three FSWs had unknown HIV status, 44 FSWs did not report the time of HIV testing, and 47 FSWs did not respond to the HIV testing questions. Overall, 44.7% ((n = 584) 95% CI: 40.7; 48.7) reported HIV testing in the last 12 months (Table 1). A higher prevalence of HIV testing in the last 12 months was reported among FSWs who were married (45.2%), had a college education (50.8%), and were 18 years old or more at first sex (46.3%). HIV testing prevalence varied by geographical location. FSWs in Shiraz (77.5%), Khorramabad (54.0%), and Tehran (49.0%) had in highest HIV testing prevalence. However, the lowest prevalence was seen in Tabriz (8.4%), and Bandar Abbas (18.3%) (Fig 1).

In the bivariable model (Table 2), age, income in the last month, group sex in the last 12 months, condom use at last sex with a paying partner, condom use at last sex with a non-paying partner, lifetime history of alcohol consumption, lifetime history of drug use, HIV knowledge, and self-perceived risk of HIV had a significant association with HIV testing in the last 12 months among FSWs. After adjusting for covariates in the multivariable model, we found that HIV testing in the last 12 months was more prevalent among FSWs who used condoms at last sex with a non-paying partner (OR = 1.78; 95% CI: 1.39; 2.27), not had a lifetime history of alcohol consumption (OR = 1.72; 95% CI: 1.33; 2.23), had a lifetime history of drug use (OR = 1.60; 95% CI: 1.22; 2.08) (Table 3).

The three most common reasons for not being tested for HIV were 1) not aware of an HIV testing site (21.2%), 2) not having enough time and not being easy HIV testing (19.5%), and 3) not considering themselves to be at risk for HIV (15.5%) (Fig 2).

## Discussion

Our findings suggested that less than half of FSWs were tested for HIV in the last 12 months, which is significantly lower than the 70.1% reported in 2015 [9]. We found that the prevalence of HIV testing in the last 12 months was slightly higher among FSWs who used a condom at last sex with a non-paying partner (56.5%), not had a lifetime history of alcohol consumption (48.3%), and had a lifetime history of drug use (51.3%). But overall, it was very low across all demographic and subgroups defined by behaviors. Several barriers to HIV testing were reported,

**Table 1. HIV testing in the last 12 months results among FSWs by different subgroups of demographics and risk behaviors in Iran, 2020.**

| Variables | Total | Number tested in the last 12 months | Unweighted prevalence (CI) | RDS weighted prevalence (CI) |
|---|---|---|---|---|
| Overall | 1399 | 584 | 41.7 (39.2; 44.3) | 44.7 (40.7; 48.7) |
| **Age (year) (Mean ± SD)** | | | 35.5 (9.0) | 35.5 (9.0) |
| **Current marital status** | | | | |
| Single | 172 | 72 | 41.9 (34.6; 49.4) | 41.6 (31.6; 51.6) |
| Married | 297 | 121 | 40.7 (35.2; 46.4) | 45.2 (37.6; 52.8) |
| Divorced, widowed, or sigheh[1] | 923 | 387 | 41.9 (38.8; 45.1) | 45.1 (40.3; 49.9) |
| **Educational level** | | | | |
| Illiterate or primary school | 379 | 158 | 41.7 (36.8; 46.7) | 43.1 (35.9; 50.1) |
| Secondary or high school | 870 | 361 | 41.5 (38.2; 44.8) | 44.4 (39.7; 49.1) |
| College education | 149 | 64 | 42.9 (35.2; 51.1) | 50.8 (39.6; 61.9) |
| Monthly income (dollars) (Mean ± SD) | | | 215.3 (222.3) | 215.3 (222.3) |
| **Age at first sex** | | | | |
| <18 | 825 | 335 | 40.6 (37.3; 44.0) | 43.4 (38.5; 48.3) |
| ≥18 | 555 | 239 | 43.1 (38.9; 47.2) | 46.3 (40.4; 52.2) |
| **Age at initiation of sex work** | | | | |
| <18 | 143 | 54 | 37.8 (30.1; 46.1) | 41.1 (30.4; 51.9) |
| ≥18 | 1182 | 504 | 42.6 (39.8; 45.5) | 46.0 (41.8; 50.3) |
| **Age at initiation of drug use** | | | | |
| <18 | 64 | 27 | 42.2 (30.5; 54.8) | 47.8 (31.5; 63.47) |
| ≥18 | 230 | 123 | 53.5 (46.9; 59.8) | 56.6 (47.9; 65.2) |
| **History of injection drug use (ever)** | | | | |
| Yes | 35 | 19 | 54.3 (37.1; 70.5) | 74.5 (55.6; 92.8) |
| No | 378 | 175 | 46.3 (41.3; 51.4) | 49.0 (42.1; 55.9) |
| **Group sex in last 12 months** | | | | |
| Yes | 273 | 88 | 32.2 (26.9; 38.0) | 40.6 (31.2; 50.1) |
| No | 121 | 27 | 22.3 (15.7; 30.7) | 30.1 (17.0; 43.1) |
| **Lifetime abortion** | | | | |
| Yes | 555 | 225 | 40.5 (36.5; 44.7) | 41.8 (36.2; 47.5) |
| No | 729 | 313 | 42.9 (39.4; 46.5) | 46.5 (41.5; 51.5) |
| Condom use at last sex with paying partner | | | | |
| Yes | 958 | 471 | 49.2 (46.0; 52.3) | 52.6 (48.2; 57.1) |
| No | 433 | 110 | 25.4 (21.5; 29.7) | 27.3 (21.7; 32.9) |
| **Condom use at last sex with non-paying partner** | | | | |
| Yes | 468 | 228 | 48.7 (44.2; 53.3) | 56.5 (49.6; 63.5) |
| No | 719 | 250 | 34.8 (31.4; 38.3) | 36.7 (31.9; 41.4) |
| History of previous incarceration in last 12 month | | | | |
| Yes | 100 | 53 | 53.0 (43.0; 62.7) | 68.9 (57.1; 80.7) |
| No | 247 | 106 | 42.9 (36.8; 49.2) | 50.5 (40.4; 60.5) |
| **Lifetime history of alcohol consumption** | | | | |
| Yes | 830 | 311 | 37.5 (34.2; 40.8) | 42.9 (38.1; 47.8) |
| No | 522 | 256 | 49.1 (44.7; 53.3) | 48.3 (42.4; 54.1) |
| **Lifetime history of drug use** | | | | |
| Yes | 443 | 211 | 47.6 (42.9; 52.3) | 51.3 (44.3; 58.2) |
| No | 923 | 361 | 39.1 (36.0; 42.3) | 42.6 (37.8; 47.4) |
| Lifetime receiving MMT* | | | | |
| Yes | 29 | 16 | 55.2 (36.1; 72.8) | 57.8 (47.9; 63.0) |
| No | 16 | 5 | 31.3 (12.2; 59.8) | 39.8 (38.1; 46.9) |

*(Continued)*

**Table 1.** (Continued)

| Variables | Total | Number tested in the last 12 months | Unweighted prevalence (CI) | RDS weighted prevalence (CI) |
|---|---|---|---|---|
| HIV knowledge[2] | | | | |
| Sufficient | 239 | 170 | 71.1 (65.0; 76.5) | 77.2 (70.6; 83.6) |
| Insufficient | 1160 | 414 | 35.7 (32.9; 38.5) | 38.7 (34.3; 43.0) |
| Self-perceived risk of HIV[3] | | | | |
| Yes | 1010 | 427 | 42.3 (39.2; 45.3) | 47.1 (42.3; 51.8) |
| No | 225 | 113 | 50.2 (43.6; 56.7) | 48.9 (40.4; 57.5) |

1) Temporary marriage

2) Sufficient knowledge about the transmission of HIV

3) The risk of contracting HIV from the individual's point of view

* Methadone maintenance therapy

such as not knowing an HIV testing site, not having enough time, not being easy to HIV testing, and not knowing themselves at risk of HIV. There was considerable variability in the prevalence of HIV testing in different cities (8.4% in Tabriz and 77.5% in Shiraz). These variations could be related to differences in access to harm reduction services and HIV testing centers.

Although HIV testing services are free in Iran, across all subgroups of FSWs, we observed a very low prevalence of recent HIV testing. We found that HIV testing dropped significantly

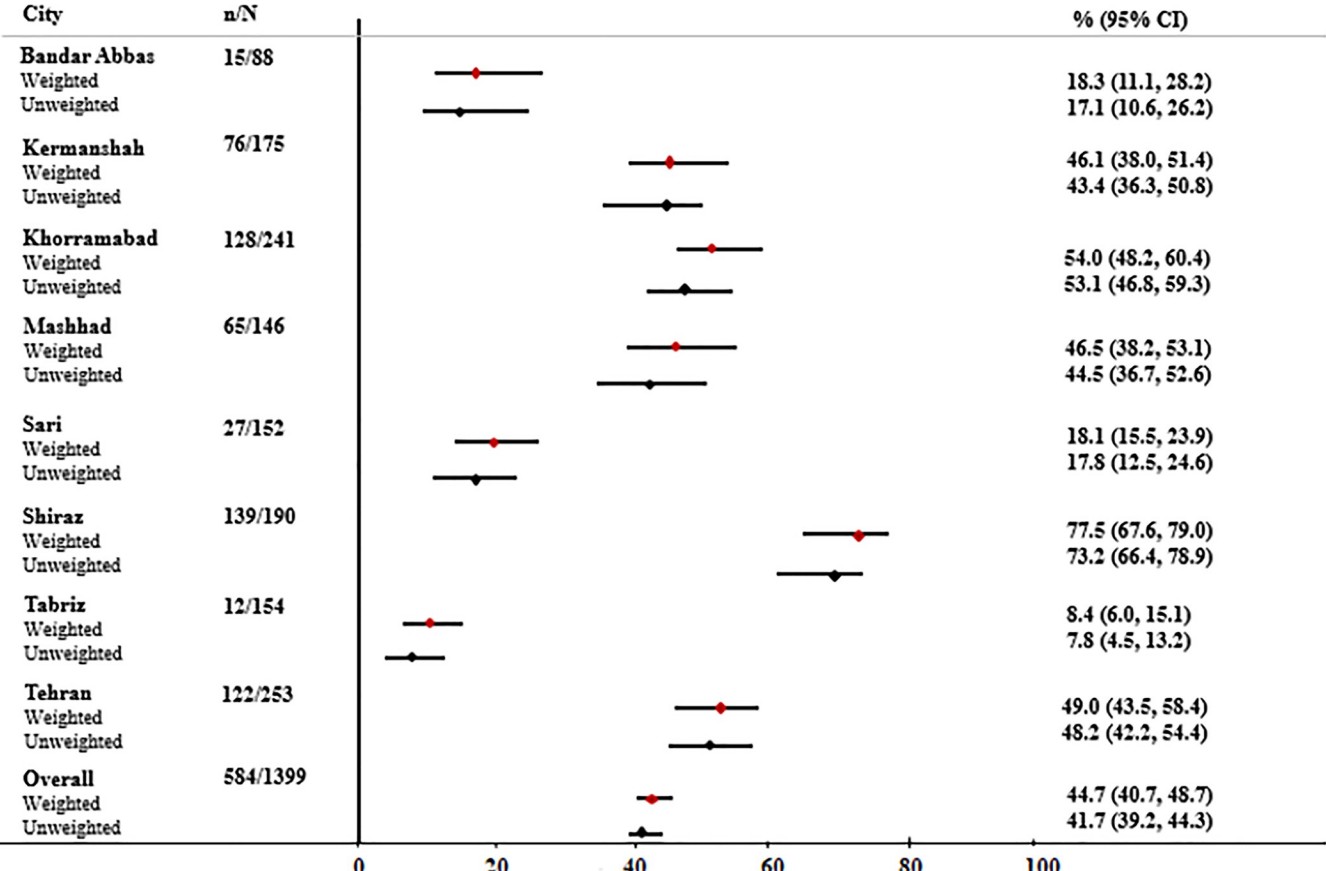

**Fig 1. HIV testing RDS weighted and unweighted prevalence among female sex workers in eight cities in Iran, 2020.**

**Table 2. Bivariable logistic regression on HIV testing in the last 12 months among FSWs in Iran, 2020.**

| Variables | HIV testing in the last 12 months[1] | |
|---|---|---|
| | Crude OR (95% CI) | P-value |
| **Age (year)** | 1.01 (1.00; 1.02) | 0.039 |
| **Current marital status** | | |
| Single | Ref | Ref |
| Married | 0.95 (0.65; 1.39) | 0.812 |
| Divorced, widowed, or sigheh[2] | 1.00 (0.72; 1.39) | 0.987 |
| **Educational level** | | |
| Illiterate or primary school | Ref | Ref |
| Secondary or high school | 0.99 (0.77; 1.26) | 0.949 |
| College education | 1.05 (0.71; 1.54) | 0.791 |
| **Monthly income (dollars)** | 0.99 (0.98; 1.00) | 0.006 |
| **Age at first sex** | | |
| <18 | Ref | Ref |
| ≥18 | 1.10 (0.88; 1.37) | 0.364 |
| **Age at initiation of sex work** | | |
| <18 | Ref | Ref |
| ≥18 | 1.22 (0.85; 1.75) | 0.265 |
| **Age at initiation of drug use** | | |
| <18 | Ref | Ref |
| ≥18 | 1.57 (0.90; 2.75) | 0.112 |
| **History of injection drug use (ever)** | | |
| No | Ref | Ref |
| Yes | 1.37 (0.68; 2.76) | 0.366 |
| **Group sex in last 12 months** | | |
| No | Ref | Ref |
| Yes | 1.65 (1.00; 2.72) | 0.047 |
| **Lifetime abortion** | | |
| No | Ref | Ref |
| Yes | 0.90 (0.72; 1.13) | 0.389 |
| **Condom use at last sex with paying partner** | | |
| No | Ref | Ref |
| Yes | 2.83 (2.21; 3.64) | <0.001 |
| **Condom use at last sex with non-paying partner** | | |
| No | Ref | Ref |
| Yes | 1.78 (1.40; 2.25) | <0.001 |
| History of previous incarceration in last 12 months | | |
| No | Ref | Ref |
| Yes | 1.50 (0.94; 2.39) | 0.089 |
| **Lifetime history of alcohol consumption** | | |
| Yes | Ref | Ref |
| No | 1.60 (1.28; 2.00) | <0.001 |
| **Lifetime history of drug use** | | |
| No | Ref | Ref |
| Yes | 1.41 (1.12; 1.77) | 0.003 |
| Lifetime receiving MMT* | | |
| No | Ref | Ref |
| Yes | 2.70 (0.74; 9.79) | 0.129 |

(*Continued*)

**Table 2.** (Continued)

| Variables | HIV testing in the last 12 months[1] | |
|---|---|---|
| HIV knowledge[3] | | |
| Insufficient | Ref | Ref |
| Sufficient | 4.43 (3.27; 6.01) | <0.001 |
| Self-perceived risk of HIV[4] | | |
| No | Ref | Ref |
| Yes | 0.72 (0.54; 0.96) | 0.030 |

1) Unweighted regression model

2) Temporary marriage

3) Sufficient knowledge about the transmission of HIV

4) The risk of contracting HIV from the individual's point of view

* Methadone maintenance therapy

among FSWs in 2020 compared to the 70.1% reported for 2015 [9]. The results of another study among FSWs in 2017 reported that about half of FSWs tested for HIV in a lifetime [10]. Some of these differences may be due to the different sampling methods for recruiting participants in these studies. For example, in the current study, we used the RDS, a peer-referral sampling methodology, to recruit FSWs. However, the 2015 FSWs survey used facility-based sampling. Recruitment from health facilities may likely have resulted in an overestimation of the HIV testing prevalence in 2015. A subgroup analysis of 2015 survey data showed that HIV testing was 34.0% (95%CI, 22.8% to 47.4%) among FSW recruited from outreach sites [9]. All these data showed that HIV testing among FSWs in Iran is still concerning low.

A slightly higher prevalence of recent HIV testing was seen among FSWs who used condoms at the last sex with a non-paying partner. While we cannot fully assess the reasons here, those who use condoms may have more access to condom services as well as HIV testing services, they might be aware of HIV prevention strategies (condom use, frequent testing, etc.), or in general, they have much more healthy behaviors such as using condom and test for HIV frequently. Data from a study in China suggested that harm reduction strategies may be an effective means of reducing unprotected sex with clients and increasing HIV testing among FSWs [18].

Our result indicated that HIV testing in the last 12 months was more prevalent among FSWs without a history of alcohol consumption. A study in Uganda also reported alcohol use

**Table 3. Multivariable logistic regression on HIV testing in the last 12 months among FSWs in Iran, 2020.**

| Variables | HIV testing in the last 12 months (n = 1131)[1,2] | |
|---|---|---|
| | Adjusted OR (%95 CI) | P-value |
| **Condom use in the last sex with non-paying partner** | | |
| No | Ref | Ref |
| Yes | 1.78 (1.39; 2.27) | <0.001 |
| **Lifetime history of alcohol consumption** | | |
| Yes | Ref | Ref |
| No | 1.72 (1.33; 2.23) | <0.001 |
| **Lifetime history of drug use** | | |
| No | Ref | Ref |
| Yes | 1.60 (1.22; 2.08) | 0.001 |

1) Controlled for the effects of the current age, educational level, and income.

2) Unweighted regression model

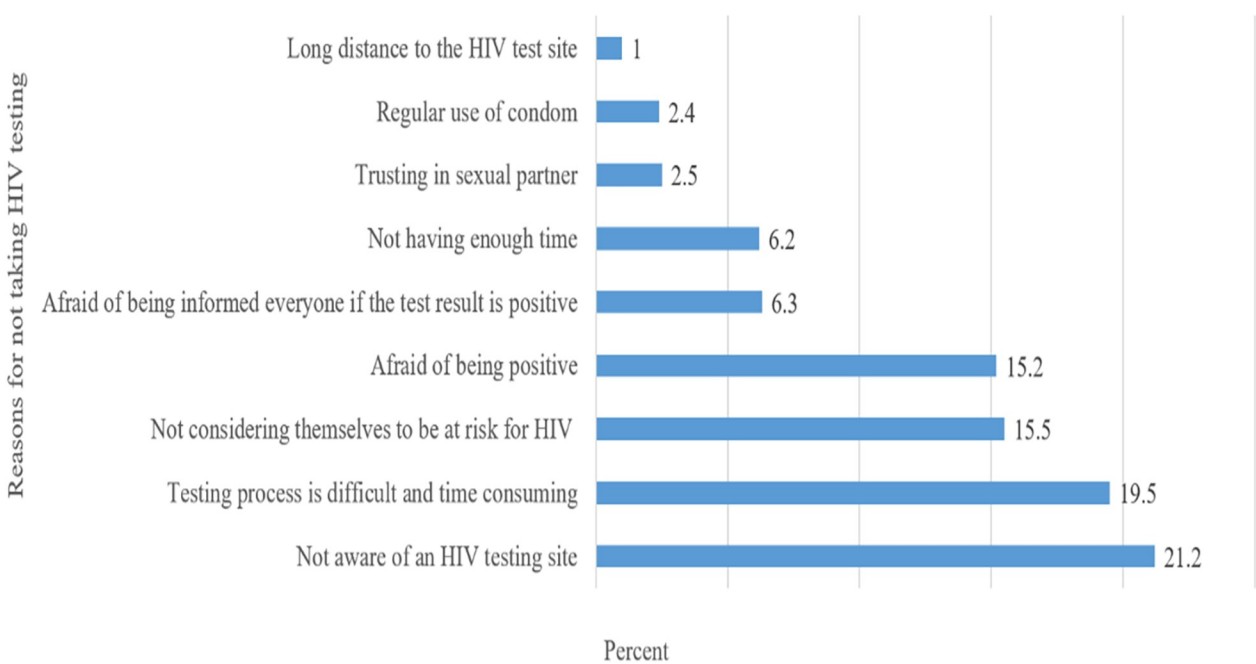

**Fig 2. Reasons for not taking HIV testing among female sex workers in Iran, 2020.**

associated with a decrease in HIV testing among women [19]. Health behaviors often tend to cluster together, so not being tested for HIV and also drinking alcohol. It is also possible that women who used alcohol could avoid HIV testing because of the stigma of alcohol use, HIV testing, and sex work [20, 21]. However, we saw a different relationship between drug use and HIV testing; FSWs who used drugs had a higher proportion of HIV testing. This is also reported in other countries like the United States, where they reported that people who use drugs were more likely to be tested for HIV [22]. It is possible that FSWs who use drugs prefer or go to facilities for drug use treatment and may have a better chance of having access to testing. Also, prevention programs in Iran focus on targeting drug injection and drug use behaviors and those FSWs who use or inject drugs [23].

We found several barriers to HIV testing. The most commonly reported barriers were not knowing a place for HIV testing, not having enough time to go to a testing site and get tested, considering HIV testing as a not easy test, and not considering themselves at risk of HIV. A recent systematic review stated that one of the main barriers to HIV testing was commonly financial and time costs, including low income, transportation costs, time constraints, and formal/informal payments [2]. Low access to testing sites and low HIV risk perception were previously reported as key barriers to the HIV testing [24].

We would like to acknowledge the four main limitations of our study. First, a face-to-face interview tends to lead to more social desirability bias, and its use may have inflated reported testing. Moreover, recall bias also could affect the prevalence of HIV testing. Second, our findings may not be generalizable to all FSWs in Iran as participants were recruited from the main cities of the most populated provinces where access to healthcare services would be better than in smaller cities or rural settings. Third, our cross-sectional study design limits causal inferences. Fourth, the study was conducted during the COVID-19 pandemic when access to our

study and other services differed from other times and conditions; for example, due to the COVID-19 pandemic, some cities collected less than the estimated sample size.

## Conclusion

We found that less than 45% of FSWs were tested for HIV in the last 12 months, which is significantly less than the reported 70% for 2015. Addressing barriers to HIV testing by using innovative and more accessible testing strategies like home-based HIV testing may address this huge gap in accessing and using HIV testing services among FSWs in Iran.

## Supporting information

**S1 File. Inclusivity in global research.**
(DOCX)

## Acknowledgments

The authors are grateful to provincial supervisors, participants, investigators, and staff for their support.

## Author Contributions

**Conceptualization:** Ghobad Moradi, Hamid Sharifi.

**Data curation:** Fatemeh Tavakoli, Bushra Zarei.

**Methodology:** Ghobad Moradi, Ali Mirzazadeh, Bushra Zarei, Hamid Sharifi.

**Project administration:** Ghobad Moradi.

**Resources:** Fatemeh Tavakoli, Ali Mirzazadeh, Bushra Zarei.

**Software:** Fatemeh Tavakoli, Bushra Zarei.

**Supervision:** Ghobad Moradi, Ali Mirzazadeh, Hamid Sharifi.

**Validation:** Ali Mirzazadeh, Bushra Zarei, Hamid Sharifi.

**Visualization:** Hamid Sharifi.

**Writing – original draft:** Fatemeh Tavakoli, Ghobad Moradi, Ali Mirzazadeh.

**Writing – review & editing:** Fatemeh Tavakoli, Ghobad Moradi, Ali Mirzazadeh, Hamid Sharifi.

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
