## [Decision Letter · Decision Letter 0]

27 Feb 2023

PONE-D-23-01726HIV Testing Dropped Significantly among Female Sex Workers in Iran in 2020PLOS ONE

Dear Dr. Sharifi,

Thank you for submitting your manuscript to PLOS ONE. After careful consideration, we feel that it has merit but does not fully meet PLOS ONE’s publication criteria as it currently stands. Therefore, we invite you to submit a revised version of the manuscript that addresses the points raised during the review process.

We look forward to receiving your revised manuscript.

Kind regards,

Orvalho Augusto, MD, MPH

Academic Editor

PLOS ONE

Journal Requirements:

2. Please include a complete copy of PLOS’ questionnaire on inclusivity in global research in your revised manuscript. Our policy for research in this area aims to improve transparency in the reporting of research performed outside of researchers’ own country or community. The policy applies to researchers who have travelled to a different country to conduct research, research with Indigenous populations or their lands, and research on cultural artefacts. 

The questionnaire can also be requested at the journal’s discretion for any other submissions, even if these conditions are not met.  Please find more information on the policy and a link to download a blank copy of the questionnaire here:

https://journals.plos.org/plosone/s/best-practices-in-research-reporting. 

Please upload a completed version of your questionnaire as Supporting Information when you resubmit your manuscript.

- https://doi.org/10.1007/s10461-022-03827-x

In your revision ensure you cite all your sources (including your own works), and quote or rephrase any duplicated text outside the methods section. Further consideration is dependent on these concerns being addressed.

"We acknowledge the support from the University of California, San Francisco’s International Traineeships in AIDS Prevention Studies (ITAPS), U.S. NIMH, R25MH123256."

6. We note that you have indicated that data from this study are available upon request. PLOS only allows data to be available upon request if there are legal or ethical restrictions on sharing data publicly. For more information on unacceptable data access restrictions, please see http://journals.plos.org/plosone/s/data-availability#loc-unacceptable-data-access-restrictions. 

7. Please include a caption for figure 1. 

**Additional Editor Comments:**

This is an important report for HIV control not only in Iran but in similar places around the globe. The authors conducted an RDS-based survey among female sex workers (FSW) to assess the prevalence/coverage of HIV testing in the last 12 months. However, the authors did a poor job of documenting critical methodologic RDS details to understand and appreciate the report as the reviewer points out below.

Specific issues:

1. As the reviewer indicate below please use the STROBE-RDS to improve the report.

2. The study design and setting currently fail to describe any setting. In the current citation number 12 there are details that are relevant to include.

3. About the independent variables why age and income are just analysed as binary variables? At least, univariate of these variables should also include them as continuous so we can learn about their mean, standard deviation, quartiles and range.

Line 103 speaks of an aggregation. Please provide details on such an aggregation was done.

4. Lines 110 to 112 speak of a procedure to select variables to adjust for. The problem here is how these p-values were calculated. Are the authors using RDS accounting procedures to get these p-values?

5. Tables 2 and 3 are these results based on an RDS-aware procedure? What software was used for the analysis? Please include that information in the statical considerations section.

6. Also, please revise the title. According to this report, this is the first RDS-based study for HIV testing among Iran women. Although it may be true that there was a drop, the previous data is based on non-comparable sampling procedures. Therefore it is misleading to speak of "dropping" here.

Reviewers' comments:

Reviewer's Responses to Questions

**Comments to the Author**

1. Is the manuscript technically sound, and do the data support the conclusions?

Reviewer #1: Yes

Reviewer #2: Yes

2. Has the statistical analysis been performed appropriately and rigorously? 

Reviewer #1: I Don't Know

Reviewer #2: Yes

3. Have the authors made all data underlying the findings in their manuscript fully available?

Reviewer #1: No

Reviewer #2: No

4. Is the manuscript presented in an intelligible fashion and written in standard English?

Reviewer #1: No

Reviewer #2: Yes

5. Review Comments to the Author

Reviewer #1: The authors used an appropriate quasi-probabilistic method to recruit a large sample of hard-to-reach population of female sex workers in eight cities in Iran. The data are not shared because of confidentiality reasons, which I find appropriate for this highly stigmatised population. The authors present findings related to HIV testing, explore correlates of HIV testing in the last 12 months and discuss the relevance of their results for the prevention effort in a comprehensive way. The importance of data on HIV testing in this population in Iran is highlighted in an appropriate way it the Introduction section. However, the overall reporting in this interesting work could be improved and aligned with the reporting standards for this study design, in particular the Methods and the Results section, so that the details of the authors’ approach and possible limitations can be better understood. The authors should also check the manuscript carefully and correct typographical and grammatical errors.

1. It would be helpful for the readers if the report included items in line with the STROBE-RDS Statement checklist (R.G. White et al. / Journal of Clinical Epidemiology 68 (2015) 1463e1471), details possibly also as a supplementary material. The overall reporting is somewhat unclear.

2. Title should explicitly include the study design: e.g. “HIV testing…Iran in 2020: findings from a respondent driven sampling surveys” and be more aligned with the content of the abstract

3. Abstract - Details of methods - biological samples + face-to-face interviews should also be stated. HIV-prevalence is usually explored in RDS, so HIV testing was probably one of the primary outcomes, and not the only one, so I suggest deleting that sentence and replacing it with “We used multivariable logistic regression to explore correlates of HIV testing in the last 12 months” to reflect the statistical approach explicitly.

4. The language of the abstract should be revised carefully and rewritten

5. Although authors cite an article where details of the methods were reported, more detail should be added in line with RDS—STROBE requirements which are relevant for the readers to better understand and interpret the multivariate analysis the authors present (e.g. formative research basic findings, potential sources of bias and efforts to address them, groupings of variables)

6. P4L72 “For the current analysis, participants were excluded if they were HIV-positive, had unknown HIV status, or did not report their HIV testing history or the

time of HIV testing.” – Why were these participants excluded? All HIV positive, or only HIV positive who were aware of their HIV status? What were the proportions of these subgroups in the sample (to be presented in the results). This is confusing when read with lines 82-90, consider presenting the process of exclusion and coding in a flow diagram for it to be clearer

7. P5L92 – there seems to be a typo for age groups, both are <29

8. The concept of “sigheh” should be briefly explained for international readership

9. P5L96-97 “age at first drug use (<18 vs. ≥18), injection drug use (ever) (yes vs. no)”, this is confusing

10. Please check that the statistical analysis section is written in line with STROBE-RDS Statement item 12

11. P6L118 Please state reasons for exclusion from the analysis (and %)

12. Table 1 Sufficient/Insufficient knowledge should be defined more precisely (in the Methods section)

13. “Self-perceived risk of HIV” – reword to perceives personal risk as high?

14. Consider merging and simplifying Table 2 and 3

15. It should be stated what was the number of participants included in the final model in multivariate analysis presented in Table 3 and in footnotes also what was controlled for

16. Why were differences across cities not explored, would you expect them and reasons for that?

17. L149 typo 56.5%

18. Figure 1 Unclear labelling, please revise

Reviewer #2: Overall this is a clear, well-structured and easy to read paper. The analysis seems sound. I have the following minor comments:

Abstract

The abstract is well-structured and clear.

I would suggest changing to ‘…was greater among FSWs who used a condom at last sex…’

‘at last sex’ should be used throughout the manuscript.

Introduction

The introduction provides a good overview of the literature.

FSW is in brackets in line 58. Remove and spell out FSW when it is first used.

I suggest adding some ambiguity to the statement that 70% of FSWs tested in 2015. As you write in the Discussion, this is likely inflated due to the recruitment method. You could also add that this study is not just an an update, but also an improvement as it uses RDS.

Methods

Are all the sampled cities listed? If so, I suggest ‘…cities in Iran, namely Bandar-…’

I understand that the sampling is described elsewhere, but perhaps add a sentence or two on seed selection etc.

Was consent requested for the interview and for tested separately? If so, were there women who didn’t consent to being tested?

Typo here: (≤29 vs. <29)

Results

The results section is very clearly written.

I suggest not reporting p-values in the text.

Discussion

The discussion is clear and thorough. I suggest adding a little more on the effect of the Covid outbreak on both the conduct of the study, and likely impact on testing uptake.

Line 180: ‘refer’, should it be prefer?

Line 182: ‘Field’ should likely be a completely different word

Line 185: ‘did not knowing’, suggest ‘were not knowing’

Line 197: suggest ‘were different from any other time and conditions’

I suggest adding the interview method to the limitations. Face-to-face interviews tend to lead to more social desirability bias, and its use may have inflated reported testing.

Tables and figures

The tables are generally very clear.

Change the N and n to be clearer. Maybe ‘Total’ instead of N. And ‘Number tested past year’ instead of n.

Figure 1 needs a label on the y-axis. I suggest rounding the percentages to the nearest whole number. Decimal places suggest a level of accuracy that your sample size doesn’t provide. This applies to all percentages reported in the paper.

6. PLOS authors have the option to publish the peer review history of their article (what does this mean?). If published, this will include your full peer review and any attached files.

Reviewer #1: No

Reviewer #2: No

---

## [Author Response · Author response to Decision Letter 0]

26 May 2023

PLOS ONE 

RE: Response to Reviewers’ comments on Manuscript ID PONE-D-23-01726

Dear Dr. Orvalho Augusto

We greatly appreciate the opportunity to revise and resubmit our manuscript entitled “HIV Testing Dropped Significantly among Female Sex Workers in Iran in 2020”. We found the reviewers’ comments helpful and constructive and have tried our best to address all of them. We have also had the paper edited to improve the flow and sentence structure and think its readability has improved. We also changed the title of the manuscript based on the comment of the first reviewer, and the editor's emphasis to “HIV Testing and its associated factors among Female Sex Workers in Iran in 2020: finding from a respondent-driven sampling survey”. Below is a point-by-point explanation of how we addressed the reviewer's concerns.

One of our co-authors (Ali Mirzazadeh) has received funding from NIMH (R25MH123256). 

Please update the funding statement for this paper.

Data are owned by the Ministry of Health of Iran and are available from the HIV/STI office located in the Ministry of Health of Iran (E-mail: aids@behdasht.gov.ir; Tel: +98(0) 21-81455055) for researchers who meet the criteria for access to confidential data. Female Sex workers are a highly stigmatized population and sex work is illegal in Iran. To protect the study population, all individual-level data are considered sensitive data. It requires that all researchers who want to work on this data submit their request to the Ministry of Health.

Please update the data availability statement for this paper.

We hope that this response will be viewed favorably, and our revised manuscript will be considered for publication in PLOS ONE. 

Sincerely,

Hamid Sharifi, PhD

Professor in Epidemiology

Journal Requirements:

Response: We checked the PLOS ONE style and modified the manuscript.

2. Please include a complete copy of PLOS’ questionnaire on inclusivity in global research in your revised manuscript. Our policy for research in this area aims to improve transparency in the reporting of research performed outside of researchers’ own country or community. The policy applies to researchers who have travelled to a different country to conduct research, research with Indigenous populations or their lands, and research on cultural artefacts. 

The questionnaire can also be requested at the journal’s discretion for any other submissions, even if these conditions are not met. Please find more information on the policy and a link to download a blank copy of the questionnaire here:

https://journals.plos.org/plosone/s/best-practices-in-research-reporting. 

Please upload a completed version of your questionnaire as Supporting Information when you resubmit your manuscript.

Response: Thanks, we completed the questionnaire and we will upload it as supporting information.

- https://doi.org/10.1007/s10461-022-03827-x

In your revision ensure you cite all your sources (including your own works), and quote or rephrase any duplicated text outside the methods section. Further consideration is dependent on these concerns being addressed.

Response: Thanks, we considered the advice.

Response: Thanks, we checked this note and made the necessary changes. 

"We acknowledge the support from the University of California, San Francisco’s International Traineeships in AIDS Prevention Studies (ITAPS), U.S. NIMH, R25MH123256."

Response: Thank you. We revised it as requested.

6. We note that you have indicated that data from this study are available upon request. PLOS only allows data to be available upon request if there are legal or ethical restrictions on sharing data publicly. For more information on unacceptable data access restrictions, please see http://journals.plos.org/plosone/s/data-availability#loc-unacceptable-data-access-restrictions. 

Response: We added the data availability section before the references section and explained it. 

7. Please include a caption for figure 1. 

Response: We added a caption for it.

Additional Editor Comments:

This is an important report for HIV control not only in Iran but in similar places around the globe. The authors conducted an RDS-based survey among female sex workers (FSW) to assess the prevalence/coverage of HIV testing in the last 12 months. However, the authors did a poor job of documenting critical methodologic RDS details to understand and appreciate the report as the reviewer points out below.

Specific issues:

1. As the reviewer indicate below please use the STROBE-RDS to improve the report.

Response: Thanks for your recommendation. We checked STROBE-RDS and modified our manuscript according to it as much as possible.

2. The study design and setting currently fail to describe any setting. In the current citation number 12 there are details that are relevant to include.

Response: Thanks, we added some details to the method section.

3. About the independent variables why age and income are just analyzed as binary variables? At least, univariate of these variables should also include them as continuous so we can learn about their mean, standard deviation, quartiles and range.

Response: Thanks for your comment. We added age and income as continuous variables to the tables.

Line 103 speaks of an aggregation. Please provide details on such an aggregation was done.

Response: Thanks for your comment. We added a sentence about the HIV knowledge variable. It reads “Ten questions were asked about HIV transmission; correct responses to all questions were coded as sufficient and otherwise as insufficient”.

4. Lines 110 to 112 speak of a procedure to select variables to adjust for. The problem here is how these p-values were calculated. Are the authors using RDS accounting procedures to get these p-values?

Response: We used unweighted regression models because in RDS-weighted regression, bias is substantial, and type-I error rates are unacceptably high. (Ref: Unweighted regression models perform better than weighted regression techniques for respondent-driven sampling data: results from a simulation study)

5. Tables 2 and 3 are these results based on an RDS-aware procedure? What software was used for the analysis? Please include that information in the statistical considerations section.

Response: Thanks for your note. We added related explanations to the statistical analysis in the method section. It reads “Given the lack of consensus on the validity of weighted regression models, unweighted regression models were performed to avoid error rate, have better coverage, increase accuracy, and avoid biased results arising from the RDS weighted analyses. The RDS unweighted regression has been supported by the growing body of literature. Despite this, we also reported RDS-adjusted estimates for HIV testing in the last 12 months results among FSWs by different subgroups of demographics and risk behaviors, which considered network size and homophily within networks. RDS-adjusted estimates for HIV prevalence were calculated in RDS-Analyst”. 

6. Also, please revise the title. According to this report, this is the first RDS-based study for HIV testing among Iran women. Although it may be true that there was a drop, the previous data is based on non-comparable sampling procedures. Therefore it is misleading to speak of "dropping" here.

Response: Thanks for your comment. We changed it to “HIV testing and its associated factors among Female Sex Workers in Iran in 2020: finding from a respondent-driven sampling survey”.

Reviewers' comments:

Reviewer's Responses to Questions

Comments to the Author

1. Is the manuscript technically sound, and do the data support the conclusions?

Reviewer #1: Yes

Reviewer #2: Yes

2. Has the statistical analysis been performed appropriately and rigorously? 

Reviewer #1: I Don't Know

Reviewer #2: Yes

3. Have the authors made all data underlying the findings in their manuscript fully available?

Reviewer #1: No

Reviewer #2: No

4. Is the manuscript presented in an intelligible fashion and written in standard English?

Reviewer #1: No

Reviewer #2: Yes

5. Review Comments to the Author

Reviewer #1: The authors used an appropriate quasi-probabilistic method to recruit a large sample of hard-to-reach population of female sex workers in eight cities in Iran. The data are not shared because of confidentiality reasons, which I find appropriate for this highly stigmatized population. The authors present findings related to HIV testing, explore correlates of HIV testing in the last 12 months and discuss the relevance of their results for the prevention effort in a comprehensive way. The importance of data on HIV testing in this population in Iran is highlighted in an appropriate way it the Introduction section. However, the overall reporting in this interesting work could be improved and aligned with the reporting standards for this study design, in particular the Methods and the Results section, so that the details of the authors’ approach and possible limitations can be better understood. The authors should also check the manuscript carefully and correct typographical and grammatical errors.

1. It would be helpful for the readers if the report included items in line with the STROBE-RDS Statement checklist (R.G. White et al. / Journal of Clinical Epidemiology 68 (2015) 1463e1471), details possibly also as a supplementary material. The overall reporting is somewhat unclear.

Response: Thanks, we checked the STROBE-RDS checklist and modified our manuscript according to it.

2. Title should explicitly include the study design: e.g. “HIV testing…Iran in 2020: findings from a respondent driven sampling surveys” and be more aligned with the content of the abstract

Response: Thanks for your note. We added the suggested phrase to the title. It reads: ““HIV testing and its associated factors among Female Sex Workers in Iran in 2020: finding from a respondent-driven sampling survey”.

3. Abstract - Details of methods - biological samples + face-to-face interviews should also be stated. HIV-prevalence is usually explored in RDS, so HIV testing was probably one of the primary outcomes, and not the only one, so I suggest deleting that sentence and replacing it with “We used multivariable logistic regression to explore correlates of HIV testing in the last 12 months” to reflect the statistical approach explicitly.

Response: Thanks for your excellent comment. We edited the abstract based on your suggestion.

4. The language of the abstract should be revised carefully and rewritten

Response: Thanks, we revised the abstract.

5. Although authors cite an article where details of the methods were reported, more detail should be added in line with RDS—STROBE requirements which are relevant for the readers to better understand and interpret the multivariate analysis the authors present (e.g. formative research basic findings, potential sources of bias and efforts to address them, groupings of variables)

Response: Thanks for your note. We added some explanations. 

6. P4L72 “For the current analysis, participants were excluded if they were HIV-positive, had unknown HIV status, or did not report their HIV testing history or the time of HIV testing.” – Why were these participants excluded? All HIV positive, or only HIV positive who were aware of their HIV status? What were the proportions of these subgroups in the sample (to be presented in the results). This is confusing when read with lines 82-90, consider presenting the process of exclusion and coding in a flow diagram for it to be clearer

Response: Thanks, since we wanted to assess the correlates of HIV testing among FSWs that were at risk of HIV but they didn’t have HIV-positive test results. In addition, the number of FSWs that were excluded was low.

We added these sentences to the text for more clarification: “Among excluded participants, 22 FSWs had HIV-positive test results before this study, three FSWs had unknown HIV status, 44 FSWs did not report the time of HIV testing, and 47 FSWs did not respond to the HIV testing questions.”

7. P5L92 – there seems to be a typo for age groups, both are <29

Response: Sorry for this typo. We fixed this.

8. The concept of “sigheh” should be briefly explained for international readership

Response: Thanks, we added a related explanation to the text. It reads “which is a type of marriage with a flexible but predetermined duration”.

9. P5L96-97 “age at first drug use (<18 vs. ≥18), injection drug use (ever) (yes vs. no)”, this is confusing

Response: Thanks for your note, we modified them. 

10. Please check that the statistical analysis section is written in line with STROBE-RDS Statement item 12

Response: Thanks for your comment. We added related statements.

11. P6L118 Please state reasons for exclusion from the analysis (and %)

Response: Thanks, as the correlations of HIV testing are important among people with negative test results and we wanted to know, among FSWs who were at risk of HIV, what factors are important to doing HIV testing. So, we need the time for HIV testing, and knowing HIV status and if these conditions are not reported, we excluded these people. As we mentioned in comment 6, we added these sentences to the text:

“Among excluded participants, 22 FSWs had HIV-positive test results before this study, three FSWs had unknown HIV status, 44 FSWs did not report the time of HIV testing, and 47 FSWs did not respond to the HIV testing questions.”

12. Table 1 Sufficient/Insufficient knowledge should be defined more precisely (in the Methods section)

Response: Thanks, we added a related explanation. It reads: “Ten questions were asked about HIV transmission; correct responses to all questions were coded as sufficient and otherwise as insufficient”.

13. “Self-perceived risk of HIV” – reword to perceives personal risk as high?

Response: Thanks, it means whether the person considers herself/himself at risk of HIV or not. Also, other studies in this setting used this term. (Ref: Correlates of HIV Testing among Female Sex Workers in Iran: Findings of a National Bio-Behavioural Surveillance Survey, Remaining Gap in HIV Testing Uptake Among Female Sex Workers in Iran)

14. Consider merging and simplifying Table 2 and 3

Response: Thanks for your note, as only three variables are in Table 3, merging them makes Table 2 so long and somehow unattractive. We preferred to keep these two tables separately. 

15. It should be stated what was the number of participants included in the final model in multivariate analysis presented in Table 3 and in footnotes also what was controlled for

Response: Thanks, the number included in the final model was 1131. Also, we added in footnote the variables that the effect of them controlled.

16. Why were differences across cities not explored, would you expect them and reasons for that?

Response: Thank you for your comment. We added the HIV testing prevalence for each city in Figure 1. We also added this comparison to the method, results, and discussion. 

In the method, it reads: “The prevalence of HIV testing in the last 12 months also was reported for each city.

In the results: “HIV testing prevalence varied by geographical location. FSWs in Shiraz (73.2%), Khorramabad (53.1%), and Tehran (48.2%) had in highest HIV testing prevalence. However, the lowest prevalence was seen in Tabriz (7.8%), and Bandar Abbas (17.1%) (Fig 1).”

In the discussion, we added the reason for the differences: “There was considerable variability in the prevalence of HIV testing in cities (7.8% in Tabriz and 73.2% in Shiraz). This variety could be related to differences in access to harm reduction services and HIV testing centers.”

17. L149 typo 56.5%

Response: Thanks, we corrected this typo.

18. Figure 1 Unclear labelling, please revise

Response: Thanks for your note, we revised the labeling of it.

Reviewer #2: Overall this is a clear, well-structured and easy to read paper. The analysis seems sound. I have the following minor comments:

Abstract

The abstract is well-structured and clear.

I would suggest changing to ‘…was greater among FSWs who used a condom at last sex…’

‘at last sex’ should be used throughout the manuscript.

Response: Thanks, we changed it based on your comment.

Introduction

The introduction provides a good overview of the literature.

FSW is in brackets in line 58. Remove and spell out FSW when it is first used.

I suggest adding some ambiguity to the statement that 70% of FSWs tested in 2015. As you write in the Discussion, this is likely inflated due to the recruitment method. You could also add that this study is not just an update, but also an improvement as it uses RDS.

Response: Thanks for your consideration. We revised the introduction in the abstract and full text based on your recommendations.

Methods

Are all the sampled cities listed? If so, I suggest ‘…cities in Iran, namely Bandar-…’

Response: Thanks. Yes, all the sampled cities are listed. We revised the text based on the comment.

I understand that the sampling is described elsewhere, but perhaps add a sentence or two on seed selection etc.

Response: Thanks. We added related descriptions. It reads: “Sampling started with a non-random selection of seeds. Seeds were selected from different networks based on several criteria: age, geographic region, and risk characteristics related to the subgroup. Finally, 45 seeds were selected, with a minimum of four and a maximum of nine seeds for each city. One of the seeds was non-generative and did not recruit anyone to the study, so she was excluded from the study (44 seeds). Also, a formative assessment was conducted based on in-depth interviews and focus group discussions with key target population members at the beginning of the study”.

Was consent requested for the interview and for tested separately? If so, were there women who didn’t consent to being tested?

Response: No, the consent was done for both interviews and testing in one stage. All of the women consent to test.

Typo here: (≤29 vs. <29)

Response: Thanks. We fixed it.

Results

The results section is very clearly written.

I suggest not reporting p-values in the text.

Response: Thanks for your comment. We omitted p-values in the text of the results.

Discussion

The discussion is clear and thorough. I suggest adding a little more on the effect of the Covid outbreak on both the conduct of the study, and likely impact on testing uptake.

Response: Thanks for your comment. We added the impact of the COVID-19 pandemic to the limitations. It reads: “The study was conducted during the COVID-19 pandemic when access to our study and other services was different from any other time and conditions; for example, due to the COVID-19 pandemic, some cities collected less than the estimated sample size”.

Line 180: ‘refer’, should it be prefer?

Response: Thanks, we fixed it.

Line 182: ‘Field’ should likely be a completely different word

Response: Thanks, we changed it to behaviors.

Line 185: ‘did not knowing’, suggest ‘were not knowing’

Response: Thanks, we changed it.

Line 197: suggest ‘were different from any other time and conditions’

Response: Thanks, we modified it.

I suggest adding the interview method to the limitations. Face-to-face interviews tend to lead to more social desirability bias, and its use may have inflated reported testing.

Response: Thanks for your note. We added this to the limitation. It reads:

“A face-to-face interview tends to lead to more social desirability bias, and its use may have inflated reported testing. Moreover, recall bias also could affect the prevalence of HIV testing.”

Tables and figures

The tables are generally very clear.

Change the N and n to be clearer. Maybe ‘Total’ instead of N. And ‘Number tested past year’ instead of n.

Response: Thanks for your note. We shifted them.

Figure 1 needs a label on the y-axis. I suggest rounding the percentages to the nearest whole number. Decimal places suggest a level of accuracy that your sample size doesn’t provide. This applies to all percentages reported in the paper.

Response: Thanks. We fixed it.

6. PLOS authors have the option to publish the peer review history of their article (what does this mean?). If published, this will include your full peer review and any attached files.

Do you want your identity to be public for this peer review? For information about this choice, including consent withdrawal, please see our Privacy Policy.

Reviewer #1: No

Reviewer #2: No

---

## [Decision Letter · Decision Letter 1]

26 Jun 2023

PONE-D-23-01726R1HIV testing and its associated factors among Female Sex Workers in Iran in 2020: finding from a respondent-driven sampling surveyPLOS ONE

Dear Dr. Sharifi,

Thank you for submitting your manuscript to PLOS ONE. After careful consideration, we feel that it has merit but does not fully meet PLOS ONE’s publication criteria as it currently stands. Therefore, we invite you to submit a revised version of the manuscript that addresses the points raised during the review process.

We look forward to receiving your revised manuscript.

Kind regards,

Orvalho Augusto, MD, MPH

Academic Editor

PLOS ONE

Journal Requirements:

Additional Editor Comments (if provided):

This manuscript has improved since the last revision. However, some shortcomings persist.

1. The "study design and setting" now has good study design details. It lacks the study settings elements.

2. In the statistical section, please cite the software used (Stata and RDS-Analyst).

3. For tables 2 and 3, please indicate in the caption whether these are weighted or unweighted analyses.

4. Figure 1, is it possible to show both weighted and unweighted prevalence?

Reviewers' comments:

Reviewer's Responses to Questions

**Comments to the Author**

1. If the authors have adequately addressed your comments raised in a previous round of review and you feel that this manuscript is now acceptable for publication, you may indicate that here to bypass the “Comments to the Author” section, enter your conflict of interest statement in the “Confidential to Editor” section, and submit your "Accept" recommendation.

Reviewer #1: All comments have been addressed

Reviewer #2: All comments have been addressed

2. Is the manuscript technically sound, and do the data support the conclusions?

Reviewer #1: Yes

Reviewer #2: Yes

3. Has the statistical analysis been performed appropriately and rigorously? 

Reviewer #1: Yes

Reviewer #2: Yes

4. Have the authors made all data underlying the findings in their manuscript fully available?

Reviewer #1: No

Reviewer #2: Yes

5. Is the manuscript presented in an intelligible fashion and written in standard English?

Reviewer #1: No

Reviewer #2: Yes

6. Review Comments to the Author

Reviewer #1: The authors seem to have addressed my comments. Overall, they added clarity to their methodological and statistical approach, which, together with additional elaboration on some points in the Discussion section increased the quality of the manuscript in my opinion. However, there are still some language unclarities in the Abstract e.g. L20 “(RDS) about”, L35 “the odds …was more” and throughout the text, so I suggest them to revise the manuscript carefully for such details and correct as appropriate.

Reviewer #2: I would like to commend the authors for addressing all of my and the other reviewer's comments.

The two sets of age and income variables now in the tables are confusing. I suggest using them as numerical variables only.

Some minor language notes:

Describe odds as 'higher' or 'lower', not 'more or 'less'.

Please add any abbreiviations to the table footnotes (I see MMT, perhaps there are others).

7. PLOS authors have the option to publish the peer review history of their article (what does this mean?). If published, this will include your full peer review and any attached files.

Reviewer #1: No

Reviewer #2: No

---

## [Author Response · Author response to Decision Letter 1]

9 Jul 2023

PLOS ONE 

RE: Response to Reviewers’ comments on Manuscript ID PONE-D-23-01726R1

HIV testing and its associated factors among Female Sex Workers in Iran in 2020: finding from a respondent-driven sampling survey

Dear Dr. Orvalho Augusto

We greatly appreciate the opportunity to revise and resubmit our manuscript entitled “HIV testing and its associated factors among Female Sex Workers in Iran in 2020: finding from a respondent-driven sampling survey”. We found the editor’s and reviewers’ comments helpful and constructive and have tried our best to address all of them. Below is a point-by-point explanation of how we addressed the reviewer's concerns.

We hope that this response will be viewed favorably, and our revised manuscript will be considered for publication in PLOS ONE. 

Sincerely,

Hamid Sharifi, PhD

Professor in Epidemiology

Journal Requirements:

Response: Thanks. According to the editor’s comment, we added two related citations for software that we used in the references list.

Additional Editor Comments (if provided):

This manuscript has improved since the last revision. However, some shortcomings persist.

1. The "study design and setting" now has good study design details. It lacks the study settings elements.

Response: Thanks, we checked this note and made the necessary changes. We added subheadings into the study design and setting section for clarification. We, also, added a sentence about study sampling. It reads: “The cities were selected according to the maximum cultural and geographical variation by the study investigators following consultations with the Ministry of Health and to represent different regions across the country”. 

2. In the statistical section, please cite the software used (Stata and RDS-Analyst).

Response: Thanks for your comment. We added related citations.

3. For tables 2 and 3, please indicate in the caption whether these are weighted or unweighted analyses.

Response: Thanks, we added this note in the caption in both tables.

4. Figure 1, is it possible to show both weighted and unweighted prevalence?

Response: Thanks for your suggestion. We calculated both of weighted and unweighted prevalence. Also, we replaced weighted prevalence for HIV testing in cities in whole of the text. 

Reviewers' comments:

Reviewer's Responses to Questions

Comments to the Author

1. If the authors have adequately addressed your comments raised in a previous round of review and you feel that this manuscript is now acceptable for publication, you may indicate that here to bypass the “Comments to the Author” section, enter your conflict of interest statement in the “Confidential to Editor” section, and submit your "Accept" recommendation.

Reviewer #1: All comments have been addressed

Reviewer #2: All comments have been addressed

2. Is the manuscript technically sound, and do the data support the conclusions?

Reviewer #1: Yes

Reviewer #2: Yes

3. Has the statistical analysis been performed appropriately and rigorously? 

Reviewer #1: Yes

Reviewer #2: Yes

4. Have the authors made all data underlying the findings in their manuscript fully available?

Reviewer #1: No

Reviewer #2: Yes

5. Is the manuscript presented in an intelligible fashion and written in standard English?

Reviewer #1: No

Reviewer #2: Yes

6. Review Comments to the Author

Reviewer #1: The authors seem to have addressed my comments. Overall, they added clarity to their methodological and statistical approach, which, together with additional elaboration on some points in the Discussion section increased the quality of the manuscript in my opinion. However, there are still some language unclarities in the Abstract e.g. L20 “(RDS) about”, L35 “the odds …was more” and throughout the text, so I suggest them to revise the manuscript carefully for such details and correct as appropriate.

Response: Thanks for your note. We edited the abstract and the manuscript based on your suggestion. For example: “RDS about” changed to “RDS for”. 

Reviewer #2: I would like to commend the authors for addressing all of my and the other reviewer's comments. The two sets of age and income variables now in the tables are confusing. I suggest using them as numerical variables only.

Response: Thanks for your comment. We omitted the categorical variables of age and income.

Some minor language notes:

Describe odds as 'higher' or 'lower', not 'more or 'less'.

Please add any abbreiviations to the table footnotes (I see MMT, perhaps there are others).

Response: Thanks. We corrected them. 

7. PLOS authors have the option to publish the peer review history of their article (what does this mean?). If published, this will include your full peer review and any attached files.

Do you want your identity to be public for this peer review? For information about this choice, including consent withdrawal, please see our Privacy Policy.

Reviewer #1: No

Reviewer #2: No

---

## [Editor Report · Decision Letter 2]

17 Jul 2023

HIV testing and its associated factors among Female Sex Workers in Iran in 2020: finding from a respondent-driven sampling survey

PONE-D-23-01726R2

Dear Dr. Sharifi,

We’re pleased to inform you that your manuscript has been judged scientifically suitable for publication and will be formally accepted for publication once it meets all outstanding technical requirements.

Kind regards,

Orvalho Augusto, MD, MPH

Academic Editor

PLOS ONE

---

## [Editor Report · Acceptance letter]

19 Jul 2023

PONE-D-23-01726R2 

HIV testing and its associated factors among Female Sex Workers in Iran in 2020: finding from a respondent-driven sampling survey 

Dear Dr. Sharifi:

I'm pleased to inform you that your manuscript has been deemed suitable for publication in PLOS ONE. Congratulations! Your manuscript is now with our production department. 

Kind regards, 

on behalf of

Dr. Orvalho Augusto 

Academic Editor

PLOS ONE